# Symptomatic Chikungunya Virus Infection and Pregnancy Outcomes: A Nested Case-Control Study in French Guiana

**DOI:** 10.3390/v14122705

**Published:** 2022-12-02

**Authors:** Célia Basurko, Najeh Hcini, Magalie Demar, Philippe Abboud, Mathieu Nacher, Gabriel Carles, Véronique Lambert, Séverine Matheus

**Affiliations:** 1Centre d’Investigation Clinique Antilles Guyane, CIC Inserm 1424, Centre Hospitalier de Cayenne, Av des Flamboyants, Cayenne 97300, French Guiana; 2Department of Obstetrics and Gynaecology, West French Guiana Hospital Center, Saint Laurent du Maroni 97320, French Guiana; 3Laboratoire Hospitalo-Universitaire de Parasitologie-Mycologie, Centre Hospitalier de Cayenne, Cayenne 97300, French Guiana; 4Infectious and Tropical Diseases Department, Centre Hospitalier de Cayenne, Cayenne 97300, French Guiana; 5Intensive Care Unit, Cayenne General Hospital, Cayenne 97300, French Guiana; 6National Reference Center for Arboviruses, Laboratoire de Virology, Institut Pasteur de Guyane, Cayenne 97300, French Guiana

**Keywords:** Chikungunya virus, French Guiana, neonatal outcomes, pregnancy, pregnancy outcomes

## Abstract

During the Chikungunya epidemic in the Caribbean and Latin America, pregnant women were affected by the virus in French Guiana. The question of the impact of the virus on pregnancy was raised because of the lack of scientific consensus and published data in the region. Thus, during the Chikungunya outbreak in French Guiana, a comparative study was set up using a cohort of pregnant women. The objective was to compare pregnancy and neonatal outcomes between pregnant women with Chikungunya virus (CHIKV) infection and pregnant women without CHIKV. Of 653 mothers included in the cohort, 246 mothers were included in the case-control study: 73 had CHIKV fever during pregnancy and 173 had neither fever nor CHIKV during pregnancy. The study did not observe any severe clinical presentation of CHIKV in the participating women. There were no intensive care unit admissions. In addition, the study showed no significant difference between the two groups with regard to pregnancy complications. However, the results showed a potential excess risk of neonatal ICU admission of the newborn when the maternal infection occurred within 7 days before delivery. These results suggest that special attention should be paid to neonates whose mothers were infected with CHIKV shortly before delivery.

## 1. Introduction

Chikungunya is a mosquito-borne viral disease first described in an outbreak in southern Tanzania in 1952–1953 [1]. The clinical presentation of this disease usually includes fever and disabling joint pains [2]. To date, there is no specific treatment or vaccine. Since 2004, the Chikungunya virus (CHIKV) has spread rapidly to the Americas, Africa, Asia, and Europe [3]. Starting in December 2013, indigenous cases of CHIKV infection were detected in the Americas: from the French West Indies, the epidemic then rapidly spread to the Caribbean and Latin America [3,4]. Indeed, from December 2013 to January 2015, the number of suspected CHIKV cases in the Caribbean islands of Martinique and Guadeloupe was estimated to be about 308,000, or 40% of the population [5]. In French Guiana, a French overseas territory in South America, the first imported cases of the disease were detected in mid-December 2013 and the first autochthonous cases in February 2014 [6]. The epidemic affected approximately 16,000 people (about 6.5% of the population) between February 2014 and October 2015, with two fatal cases [7].

The clinical presentation of CHIKV infection during pregnancy generally does not seem to differ from the usual presentation [8], but there are a few published reports of severe forms [9,10,11]. In Grenada, maternal CHIKV infection was associated with a different symptomatology (more muscle and bone pain) than infection in non-pregnant women [10].A Colombian study reported cases of maternal CHIKV sepsis with admission to an intensive care unit of women infected late in pregnancy [9]. However, apart from these reports, maternal infection does not appear to be associated with major adverse effects during pregnancy [8,10]. Nonetheless, some authors have reported an increase in pregnancy morbidity and fetal mortality [12] and a risk of neonatal infection by vertical transmission during delivery. During the major epidemic that occurred in the Reunion Island (French overseas territory in the Indian Ocean) in 2005–2006, cases of mother-to-child transmission of the virus at birth and neurological complications in newborns (encephalopathy, cerebral edema, and cerebral hemorrhage) were reported [13,14].

These discrepancies likely reflect the lack of robust data in the scientific literature and emphasize that the impact of CHIKV during pregnancy still needs to be clarified [15]. Thus, the present nested case control study proposes: (i) to describe CHIKV infection in pregnant women during the 2014–2015 epidemic that affected French Guiana, and (ii) to compare adverse pregnancy and neonatal outcomes between a group of women infected with CHIKV during pregnancy and a group of control women without fever or CHIKV infection during pregnancy. The goal of this work is to provide additional information from a rigorous comparative study to help improve maternal and neonatal management of CHIKV during pregnancy.

## 2. Materials and Methods

### 2.1. Study Overview

This was a case-control study (pregCHIKV study) nested within the CMFdeng cohort [16]. Cases were women who had CHIKV infection (CHIKV group) during pregnancy and controls were women who did not have CHIKV infection (control group) during pregnancy.

Between June 2012 and June 2015, 653 pregnant women followed in French Guiana were included in the CMFdeng cohort (Figure 1. Study Flow Chart). Among them, 363 presented fever during pregnancy. These women were included in the CMFdeng cohort at the time of their fever and followed until delivery. This fever was due to dengue virus infection for 82 women and CHIKV infection for 78 women. Of these 78 women with confirmed CHIKV infection during pregnancy, 3 presented with asymptomatic CHIKV infection (fever related to another infection) and 2 were lost to follow-up (moving for delivery). With the removal of these 5 patients, 73 women were therefore included in the CHIKV group of the pregCHIKV study.

In the CMFdeng cohort, 290 women did not develop fever between the beginning of pregnancy and delivery and were included in the unexposed group [16]. Of these, 219 were free of dengue virus infection (biologically confirmed). Of these 219 women, 54 women were free of CHIKV infection (biologically confirmed) and 121 women had delivered before the first CHIKV cases in the Caribbean area (before December 2013). By removing 2 patients excluded for lack of medical data, 173 women were thus included in the control group.

### 2.2. Definition of Groups

To be included in the pregCHIKV study, cases had to have the laboratory-confirmed symptomatic Chikungunya virus infection during pregnancy. Thus, included women had to have fever and detection of the viral RNA genome by reverse transcriptase-polymerase chain reaction (RT-PCR) or seroconversionIgM/IgG antibodies in the collected samples. 

To be included in the control group, women had to have no fever, no dengue infection, and no CHIKVinfection at any time during pregnancy. Pregnant women were unexposed to CHIKV if they had given birth before the CHIK epidemic in the Caribbean area (before December 2013) or if specific IgM and IgG anti-CHIKV antibodies by “in-house”enzyme-linked immunosorbent assay testson blood sample at delivery were negative. 

### 2.3. Biological Analysis

During the epidemic, biological diagnosis was performed by the French National Reference Center for arboviruses in French Guiana. For serological diagnosis, both IgM and IgG anti-CHIKV-specific antibodies were screened in sera using an “in-house” enzyme-linked immunosorbent assay (IgM antibody capture ELISA and ELISA, respectively) as described by Talarmin et al. [17]. For the molecular investigation, viral RNA was extracted from each biological sample with the QIAamp^®^ Viral RNA kit (Qiagen, Hilden, Germany). Finally, extracted RNA samples were analyzed by real-time RT-PCR targeting the NSP1 gene of CHIKV, as described by Panning et al. [18]. Classification of patients as having laboratory-confirmed cases required either a positive RT-PCR, IgM/IgG seroconversion, or an isolated IgG detection with, at least, a subsequent 4-fold increase in titer between the first collected sample and the one obtained at delivery.

### 2.4. Data Collection and Outcome

Data were collected from medical records and at the patient’s bedside (data about the standard of care). The quality of pregnancy follow-up was defined by having a first ultrasound before 14 weeks of gestation (WG) and at least 7 prenatal visits.

The pregnancy outcomes were collected are defined in Appendix A. Given the risks of insufficient power for single pathological events, different pregnancy adverse outcomes were grouped into a single composite end point. Specifically, the criterion consisted of: preterm labor, preterm delivery, intra-uterine growth retardation, post-partum hemorrhage, caesarean section for abnormal fetal heart rate, or stillbirth.

The neonatal outcomes were collected (Appendix A) from birth to discharge (immediate postpartum). Similarly to what was done for pregnancy outcomes, different neonatal adverse outcomes were grouped into a single composite end-point. Specifically, the criterion consisted of: five-minutes Apgar score < 7, respiratory distress, seizures, hyperthermia at birth, congenital anomalies, or neonatal ICU admission.

### 2.5. Statistical Analysis

STATA 16.0 (STATA Corporation, College Station, TX, USA) was used.

First, the analysis plan consisted of checking the comparability of the 2 groups (CHIKV group versus Control group) with respect to socio-demographic characteristics, quality of pregnancy follow-up, and history using Pearson’s chi2 or Fisher’s exact tests. 

Then, pregnancy and neonatal outcomes were compared between the control group and the CHIKV group using Pearson’s chi2 or Fisher’s exact tests. The adjustment for potential confounders was performed using unconditional logistic regression. Adjustment factors were selected based on the scientific literature. The adjustment variables for the model studying the association between adverse events during pregnancy and maternal CHIKV infection were obstetrical history, primigravida, and quality of pregnancy follow-up. For the study of the association between neonatal outcomes and CHIKV infection, the adjustment variables selected were gestational age at delivery, mode of delivery, and primigravida.

Finally, for exploratory purposes, we studied pregnancy and neonatal outcomes only in the CHIKV group according to the time between fever and delivery. Thus, the CHIKV group was divided into 2 subgroups: one group of women who had CHIKV fever within 7 days before delivery and another group where women had CHIKV fever before 7 days before delivery. The 7-day threshold was chosen based on the duration of CHIKV viremia [2].

The initial significance level was 5%; *p* values were corrected for multiple testing using the Bonferroni method.

### 2.6. Ethics Statement

Written informed consent was provided and signed by all subjects before enrolment. For minors, written consent was signed by both the participant and her legal representative. The study was approved by the French regulatory authorities CCTIRS (Advisory Committee on Information Processing in Material Research in the Field of Health n° 12323), the CNIL (National Commission on Informatics and Liberty; authorization DR-2012-585), and the CEEI (Inserm’s ethics committee/Institutional Review Board, approval n° 00003888).

## 3. Results

### 3.1. Participant Characteristics 

In this case-control study, 246 pregnant women were enrolled: 73 cases in the CHIKV group and 173 in the control group. Among the CHIKV group, biological confirmation was done by detection of CHIKV viral RNA genome for 42 women (56.5%) and by IgM/IgG seroconversion for 31 women (42.5%). Virus sequencing was not performed in patients with a positive CHIKV RT-PCR.

Socio-demographic characteristics, obstetrical and medical history, and quality of pregnancy follow-up did not differ significantly between the two groups (Table 1). 

### 3.2. Description of Maternal CHIKV 

Maternal CHIKV symptoms appeared in the third trimester in 63% of cases, in the second trimester in 33% of cases, and in the first trimester in 4% of cases. The median term of CHIKV infection was 30.7 weeks (IQR = 8.2). The main symptoms reported were: fever (100%), joint pain (63.0%), headache/retro-orbital pain (35.6%) (Table 2). Joint pain was more frequently reported for maternal infections occurring in the second trimester than for infections occurring in other trimesters of pregnancy (*p*-value = 0.023). The hospitalization rate for maternal CHIKV was 65.7%.Hospitalization occurred within 24 h of symptom onset in 60% of cases and between 24 and 48 h in 32% of cases. Length of hospital stay for CHIKV fever was between 1 and 3 days for 75% of women. No women were transferred to the intensive care unit (ICU).

### 3.3. Pregnancy Outcomes 

The mean gestational age was 38.5weeks (±2.1) in the CHIKV group versus 38.9 weeks (±1.8) in the control group (*p* = 0.1232). Most deliveries were vaginal (79.4 % in CHIKV group versus 75.7% in control group; *p* = 0.527). Most fetal presentations were cephalic (97.3 % in CHIKV group versus 97.1% in control group). The median duration of maternal hospitalization for vaginal delivery was 3.5 days (IQR = 1) in the CHIKV group versus 3 days (IQR = 1) in the control group (*p* = 0.2367). The median duration of maternal hospitalization for cesarean delivery was 5 days (IQR = 2) in both groups (*p* = 0.630). 

The occurrence of pregnancy adverse outcomes did not differ significantly between the two groups (Table 3). After adjustment for obstetrical history, being a primigravida, and quality of pregnancy follow-up, the association between pregnancy adverse outcomes (composite variable) and CHIKV during pregnancy (group variable) was still not significant (Wald test *p* = 0.361, AOR = 1.33 [0.72–2.43]).

### 3.4. Neonatal Outcomes

Of 246 pregnancies, 114 newborns were male (46.3%). The mean birth weight of newborns was not different between the 2 groups: 3176 g (+/− 556 g) in the control group versus 3122 g (+/− 526) in the CHIKV group (*p* = 0.480). After excluding scheduled cesarean section (25/246), labor was spontaneous in 71.4% of cases for the CHIKV group and in 79.1% of cases for the control group (*p* = 0.221). 

Congenital anomalies detected were: polymalformative syndrome (1), renal anomalies (1), undescended testis (3), Down syndrome (1), omphalocele (1), clubfoot (5), and cutaneous lesions (4). 

The frequency of neonatal outcomes was not significantly different between the two groups (Table 4). After adjustment for preterm delivery, being a primigravida, and mode of delivery, the association between neonatal adverse outcomes (composite variable) and CHIKV during pregnancy (group variable) was still not significant (Wald test *p* = 0.327, AOR = 1.45 [0.69–3.08])

The status of the newborn at birth in the CHIKV group according to the time from infection to delivery (>7 days and ≤7 days) is presented in Table 5. It shows that neonatal ICU admission seemed more frequent among women having had CHIKV infection within 1 week before delivery than in those with earlier infection.

## 4. Discussion

In this carefully designed nested case-control study, the two groups were similar with respect to socioeconomic characteristics, obstetrical history, and medical follow-up. Considering pregnancy, delivery, or the neonatal period, we show that, overall, CHIKV infection during pregnancy was not associated with any significant difference in outcomes compared with controls. In addition, we observed that admission to the neonatal ICU appeared to be greater in mothers infected within 7 days of delivery than in mothers infected earlier in pregnancy. 

As other studies, the present study may lack statistical power because of the small number of positive CHIKV cases. It, however, contributes additional information on a specific subject which is still underexplored. Although limited by its retrospective design, the collection of data and samples was similar in the two groups. In addition, the risk of underreporting is mainly related to minor or non-severe symptoms and not to major or severe events. Despite the limitations mentioned above, two methodological features should be highlighted in this study. First, because the study was nested in a cohort, inclusion of participants was done during pregnancy, which allowed for more reliable data collection. Second, strict biological and clinical criteria were used to define the comparison groups. Indeed, the group of women infected during pregnancy were required to have had at least one fever and biological confirmation of CHIKV infection; the presence of only positive IgM for CHIKV was not sufficient to prove infection during pregnancy [19] and the diagnosis had to be confirmed either by seroconversion on two successive samples during pregnancy or by a positive RT-PCR result. Women in the control group had to either have given birth before the arrival of CHIKV in the Caribbean area, or to have negative IgM/IgG serology at delivery. The epidemic dynamic in French Guiana and the arrival of CHIKV were well documented by the National Reference Center for Arboviroses, Institut Pasteur de la Guyane) who began close surveillance of when CHIKV emergence began in January 2013 in the general population [6].

During the 2014–2015 epidemic, the majority of patients hospitalized for CHIKV infection in French Guiana presented a typical clinical picture. A few atypical or severe cases were reported, including neurological forms [7]. In our study, no atypical or severe cases were reported during pregnancy. The clinical presentation did not differ from that observed in non-pregnant patients [9,10,20]. Indeed, the three most frequently reported symptoms, regardless of the trimester of infection, were fever, joint pain, and headache/retro-orbital pain [4,6,13]. It appears that joint pain was more frequently reported when the infection occurred in the second trimester. Given the small sample size in this subgroup analysis, larger studies are needed to confirm these observations. Few severe cases of maternal CHIKV infection requiring intensive care admission have been reported in the literature [6]. Perhaps these publications are rare exceptions to the rule or result from the effect of different sociodemographic and maternal contexts (history, immunity, pregnancy follow-up, etc.).

The study in French Guiana confirmed the results observed in other comparative studies of Chikungunya infection during pregnancy [8,10,21]. Maternal CHIKV did not increase the risk of pathological pregnancy. The comparability of the groups in terms of sociodemographic characteristics, obstetrical history, and pregnancy follow-up allowed us to manage potential confounders. 

The study compared clinical and non-biological endpoints, particularly in neonates. No biological information regarding vertical transmission of CHIKV was available, which is a limitation of the present study. However, events and neonatal management were not different between cases and controls (Table 4). It is notable, however, that neonatal ICU admission appeared to be more frequent when maternal infection occurred shortly before delivery, in the period of maternal viremia (Table 5—significant difference).Although, this result should be interpreted with caution because of the lack of statistical power, it echoes similar observations reported in the literature. The study of vertical transmission of CHIKV during the 2005–2006 epidemic in the Reunion Island also suggested an increased risk of vertical transmission of CHIKV from mother to child during the period of maternal intrapartum viremia with a risk of severe neonatal infection (encephalopathy, hemorrhagic fever) [13,14]. Indeed, the transmission rate of CHIKV from mother to child reaches 50% when delivery occurs during the viremic period [14,22,23]. Neonatal meningoencephalitis in the context of maternal infection with CHIKV close to delivery has also been reported in French Polynesia [13] and in Latin America (El Salvador, Colombia, Dominican Republic) [24,25]. In our study, as in the West Indies, no cases of encephalopathy were observed [5,25]. One of the hypotheses put forward to explain these differences in neonatal infection is the viral genotype: the Asian isolate at the origin of the epidemics in the West Indies and French Guiana could be less virulent than the one from the Indian Ocean [26,27].

## 5. Conclusions

While the impact of CHIKV during pregnancy does not appear to increase the risk of maternal and fetal complications, vigilance is warranted when delivery occurs during the maternal viremia period. Indeed, the results of this study suggest an increased risk of admission to intensive care for the newborn when maternal CHIKV infection occurs within 7 days before delivery, i.e., during the period of viremia.

Further studies and meta-analyses of published results could focus on women infected in the week prior to delivery to estimate vertical transmission of CHIKVand newborn health status with greater statistical power.

## Figures and Tables

**Figure 1 viruses-14-02705-f001:**
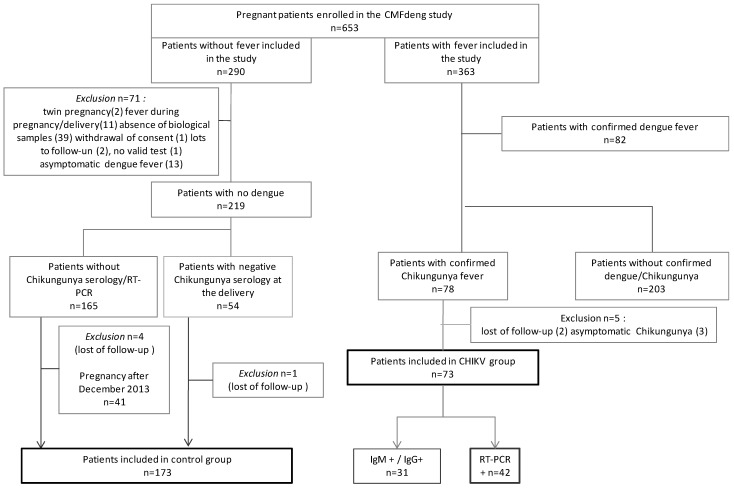
Study flow chart, pregCHIKV study 2022 [16].

**Table 1 viruses-14-02705-t001:** Socio-demographic characteristics, obstetrical and medical history, quality of pregnancy follow-up (pregCHIKV study *N* = 246).

Socio-Demographic Characteristics			
	Control Group *N* (%)	CHIKV Group *N* (%)	*p*-Value
Age at inclusion (*n =* 246)			
≤25	59 (34.1)	25 (34.2)	0.383
26–35	90 (52.0)	33 (45.2)	
≥36	24 (13.9)	15 (20.5)	
Has a complementary health insurance (*n =* 200)	119 (86.9)	57 (90.5)	0.465
Married couples (*n =* 200)	95 (69.3)	40 (63.5)	0.412
Homeowner (*n =* 200)	33 (24.1)	18 (28.6)	0.499
Had contact with family in the past 6 months (*n =* 199)	115 (84.6)	51 (80.9)	0.525
**Medical History**			
Hypertension (*n =* 245)	4 (2.3)	3 (4.2)	0.422
Diabetes (*n =* 245)	0 (0)	1 (1.4)	0.294
**Habits**			
Active smoking during pregnancy (*n =* 246)	6 (3.5)	4 (5.5)	0.489
Known alcoholism (*n =* 246)	1 (0.6)	3 (4.1)	0.080
**Gravidity upon Inclusion**			
Primigravida (*n =* 245)	37 (21.5)	9 (12.3)	0.092
**Obstetrical History (Exclusion of Primigravida Women)**			
Preterm delivery (*n =* 198)	20 (14.8)	11 (17.5)	0.633
Fetal growth restriction (*n =* 198)	7 (5.2)	3 (4.8)	1
Hypertensive disorders (*n =* 198)	17 (12.6)	11 (17.5)	0.360
Post-partum hemorrhage (*n =* 198)	8 (5.9)	3 (4.8)	1
Gestational diabetes (*n =* 198)	5 (3.7)	4 (6.3)	0.469
Induced abortions (*n =* 198)	44 (32.6)	22 (34.9)	0.746
Medical termination of pregnancy (*n =* 199)	6 (4.4)	3 (4.8)	1
Congenital anomalies(*n =* 198)	4 (2.9)	3 (4.8)	0.680
Miscarriage (<22WG) (*n =* 199)	44 (32.3)	19 (30.2)	0.757
Intra-uterine fetal death (*n =* 198)	6 (4.4)	1 (1.6)	0.434
Cesarean section (*n =* 199)	25 (18.4)	15 (23.8)	0.374
**Quality of Pregnancy Follow-Up**			
First ultrasound before 14 WG (*n =* 218)	127 (77.0)	37 (69.8)	0.294
At least 7 prenatal visits (*n =* 232)	122 (74.4)	51 (75.0)	0.923

**Table 2 viruses-14-02705-t002:** Main symptoms of maternal CHIKV according to the trimester of pregnancy (pregCHIKV study *N* = 73).

	Trimester of CHIKV Maternal Symptoms		
Maternal Symptoms	1st [*n* = 3] *N* (%)	2nd [*n* = 24] *N* (%)	3rd [*n* = 46] *N* (%)
Fever	3 (100)	24 (100)	46 (100)
Joint pain	2 (66.7)	20 (83.3)	24 (52.2)
Headache, retro-orbital pain	1 (33.3)	10 (41.7)	15 (32.6)
Muscle pain	0 (0)	8 (33.3)	11 (23.9)
Digestive symptoms (nausea, vomiting, pain)	1 (33.3)	7 (29.2)	5 (10.9)
Altered general condition, asthenia	0 (0)	2 (8.3)	6 (13.0)
General body pain	1 (33.3)	1 (4.2)	5 (10.9)
Chills	0 (0)	3 (12.5)	2 (4.3)
Skin rash, pruritus	1 (33.3)	2 (8.3)	2 (4.3)

**Table 3 viruses-14-02705-t003:** Frequency and comparison of pregnancy outcomes during pregnancy and delivery (pregCHIKV study *N*=246).

Occurrence of	Control Group *N* (%)	CHIKV Group *N* (%)	*p*-Value *
Pregnancy adverse outcomes (composite) (*n =* 246)	53 (30.6)	27 (37.0)	0.331
Preterm labor (*n =* 246)	17 (9.8)	6 (8.2)	0.692
Preterm delivery under 37 WG (*n =* 246)	19 (11.0)	7 (9.6)	0.745
Fetal growth restriction (*n =* 246)	11 (6.4)	4 (5.5)	0.792
Stillbirth (*n =* 246)	0 (0)	0 (0)	/
Postpartum hemorrhage (*n =* 246)	11 (6.4)	10 (13.7)	0.060
Caesarean section for abnormal fetal heart rate (*n =* 246)	14 (8.1)	7 (9.6)	0.701

* Bonferroni-adjusted *p*-value = 0.008 cutoff adjusting alpha of 5% for 6 tests.

**Table 4 viruses-14-02705-t004:** Frequency and comparison of neonatal outcomes (pregCHIKV study 2022 *N* = 246).

	Control Group *N* (%)	CHIKV Group *N* (%)	*p*-Value *
Macrosomia (*n =* 246)	9 (5.2)	2 (2.7)	0.514
Small for gestational age infants (*n =* 246)	16 (9.2)	7 (9.6)	0.933
Length of hospital stay (newborn) > 3 days (*n =* 244)	63 (36.8)	29 (39.7)	0.670
Neonatal adverse outcomes (composite) (*n =* 246)	31 (17.9)	15 (20.5)	0.629
Congenital anomalies (*n =* 246)	11/173 (4.6)	5/73 (5.5)	0.542
Five-minutes APGAR score < 7 (*n =* 244)	2/171 (1.2)	1/73 (1.4)	1
Respiratory distress (*n =* 244)	10/173 (5.8)	7/71 (9.9)	0.274
Seizures (*n =* 246)	1/173 (0.6)	0/73	1
NICU admission (*n =* 246)	21/173 (12.1)	10/73 (13.7)	0.736
Hyperthermia at birth (≥38 °C) (*n =* 183)	2/128 (1.6)	1/55 (1.8)	1

* Bonferroni-adjusted *p*-value = 0.0083 cutoff adjusting alpha of 5% for 6 tests.

**Table 5 viruses-14-02705-t005:** Frequency and comparison of pathological events in the CHIKV group according to time from maternal CHIKV symptoms to delivery (pregCHIKV study 2022 *N* = 73).

	Delay between CHIKV Maternal Symptoms and Delivery		*p*-Value
Outcomes	>7 Days *N* (%)	≤7 Days *N* (%)	
Preterm delivery under 37 WG (*n =* 73)	5 (7.6)	2 (28.6)	0.132
Preterm delivery under 34 WG (*n =* 73)	1 (1.5)	1 (14.3)	0.184
Caesarean section for abnormal fetal heart rate (*n =* 73)	6 (9.1)	1 (14.3)	0.522
Post-partum hemorrhage (*n =* 73)	9 (13.6)	1 (14.3)	1
Length of hospital stay (newborn) > 3 d. (*n =* 73)	25 (37.9)	4 (57.1)	0.425
Five-minute APGAR score < 7 (*n =* 73)	1 (1.5)	0 (0)	1
Respiratory distress (*n =* 71)	5 (7.8)	2 (28.6)	0.138
Neonatal ICU admission (*n =* 73)	7 (10.6)	3 (42.9)	0.049
Hyperthermia at birth (*n =* 55)	0 (0)	1 (20.0)	0.091

## Data Availability

Not applicable.

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
