# Peer review of "Symptomatic Chikungunya Virus Infection and Pregnancy Outcomes: A Nested Case-Control Study in French Guiana"

_viruses, 2022, doi:10.3390/v14122705_

Round 1
Reviewer 1 Report
The manuscript describes an interesting study which aims to examine the impact of CHIKV on pregnancy outcomes. The authors used a nested case_control study design on a cohort of pregnant women during an outbreak of chikungunya in French Guiana, and compared pregnancy and neonatal outcomes between pregnant women with and without chikungunya during pregnancy. The study did not show any significant differences between the 2 groupes, but the results suggest an increased risk of ICU admission of neonates born within 7 days after maternal infection.
There are several studies published in the litterature that examined the impact of CHIKV on pregnancy. Nevertheless this manuscrit is still relevant and contributes further to the knowledge of the impact of CHIKV , and in particular of the Asian American lineage. The findings are of interest for a readership of physicians and virologists, and for pregnant women who can be reassured with regard to the risk of chikungunya on their pregnancy.
The study is scientifically sound, the methodology is appropriate and the limitations of the retrospective design are clearly discussed. The methodology and results are presented in a well-structured manner and clearly dscribed.
In the conclusion, the authors state that the risk of vertical transmission of CHIKV could be greater when delivery occurs during the maternal viremia period, with a risk of complications for the newborn. This statement is however not a finding of this study, since there was no biological information regarding vertical transmission of CHIKV available. The results of the study suggest an increased risk of ICU admission (very small numbers), and the authors are right to hypothise that the need for ICU admission can be linked to vertical transmission, but they can do so on the basis of findings of published studies presenting evidence of vertical transmission. I suggest to rephrase the conclusion slightly so that it is clear what the findings of this study are (potentially increased risk of ICU admission) and what is based on the litterature (vertical transmission).
Reviewer 2 Report
The authors present a retrospective study of pregnant women who contracted chikungunya virus during pregnancy. While the science behind the manuscript is sound, there are some issues with the manuscript and some clarifications regarding the data that will improve the paper.
1. There are inconsistencies throughout the text for chikungunya please correct CHIK to CHIKV.
2. There are numerous places throughout the text where spacing is needed between words.
3. Table 1: were any other “habits” evaluated such as alcohol or drug use? Please specify.
4. Tables 3 and 4: Please delete the word “group” from the heading
5. Tables 1, 3, 4, 5: n/N(%) should be N(%)
6. Line 51: please provide citations for the sentence ending with “…..severe forms.”
7. Figure 1: patients without negative serology should read patients with positive serology.
8. Line 97: delete the crossed-out word
9. Line 102: please specify which tests were performed.
10. Line 119: please define WG
11. Lines 127-129: please provide justification for your choice of neonatal criteria. There are multiple documents neonatal CHIKV symptoms such as hyperpigmentation, cutaneous lesions, CNS issues, lethargy, poor feeding hypotonia etc. Did the infants present with other CHIKV symptoms or were they not included in the analysis? If they were omitted, they must be included.
12. Do you look for CHIKV co-infections with the patients with confirmed dengue?
13. You report PCR positive CHIKV patients. Please provide data on which genotype(s) of CHIKV was isolated from your patients. A significant body of research has linked specific genotypes and lineages of CHIKV with congenital/perinatal pathology and there is evidence to support that certain lineages and genotypes are not as pathogenic to the fetus/neonate.
14. Line 173 Capitalize (table 2)
15. The entire results section needs to ensure that p-values are provided after stating a result. For example lines 175-177. Was the rate of hospitalization significant?
a. Additional example: Lines 182-183 was please provide a p-value for gestational age.
16. Congenital abnormalities should include hyperpigmentation and cutaneous lesions. Please state which abnormalities were found in which group. Renal anomalies have been reported in CHIKV infected neonates.
17. It is likely that the CHIKV genotype circulating in French Guiana during the time of the study is not a genotype associated with congenital/perinatal pathology.
Round 2
Reviewer 2 Report
The Authors did an excellent job addressing my comments.